# Modifications of Rect-Spring to Enhance the Engagement of Ectopically Entrapped Molars with 2 Case Reports

**DOI:** 10.3390/children8090823

**Published:** 2021-09-19

**Authors:** Min Sun Song, Chung-Min Kang, Je Seon Song, Hyung-Jun Choi, Jaeho Lee, Seong-Oh Kim

**Affiliations:** Department of Pediatric Dentistry, College of Dentistry, Yonsei University, Seoul 03722, Korea; sms202@yuhs.ac (M.S.S.); kangcm@yuhs.ac (C.-M.K.); songjs@yuhs.ac (J.S.S.); choihj88@yuhs.ac (H.-J.C.); leejh@yuhs.ac (J.L.)

**Keywords:** ectopic eruption, first molar, tilted molar, distalization, obtuse angle

## Abstract

The Rect-spring appliance, used for the management of ectopically erupting molars, shows weak retention on mesially tilted molars. We present three modifications of the appliance for better engagement and their advantages. We describe cases of two 7-year-old patients with ectopically erupting maxillary first molars with a 2.2 mm and 2.5 mm depth of entrapment, respectively. The modified Rect-spring (mRS) was inserted between the ectopically erupting first molar and adjacent primary second molar, and exerted a distalization force with an interproximal wedging effect at the same time. After 3 months, the ectopically erupting first molars were successfully brought into proper occlusion. No discomfort was reported. The mRS is suitable for various locking cases except for severely tilted molars without requiring any laboratory procedures. We suggest it as the first choice for unlocking the first molars.

## 1. Introduction

Ectopic eruption of the permanent molars is a common eruption disturbance with an incidence of 4% [1,2]. It is classified into two types; irreversible and reversible [3]. According to a previous study, approximately 66% of ectopically erupting permanent molars showed self-improvement by the age of seven [4]. Left uncorrected ectopically erupting molars between the ages of seven and eight are considered irreversible ectopic eruptions [5,6]. Advanced root resorption resulting in premature exfoliation of the adjacent primary molars, space problems in the dental arches, and disturbed eruption of succedaneous teeth can be possible consequences of untreated, irreversible ectopic eruptions [7,8].

Rect-spring (RS) was introduced in 2014 for the effective treatment of ectopically erupting molars. It was made of a standard 0.9 mm stainless steel wire by chairside, manually, not ready-made, comprising two engaging mesial arms and a single occlusal helical spring [9]. Despite its simplicity, the device sometimes showed unstable engagement, mainly in mesially tilted molars. This limitation was due to the innate nature of the perpendicular angles between the arms and the occlusal spring (Figure 1A,C,E). Enlogating the vertical components to place the tip of the engaging arms underneath the undercut area of the ectopic molar was first suggested in order to circumvent this limitation; however, the extended arms caused significant discomfort.

This report presents a modified approach for deep and stable engagement without increasing the actual vertical arm length, describes two cases of ectopic eruption managed using a modified Rect-spring (mRS), and demonstrates its advantages and limitations.

## 2. Material and Methods

### 2.1. Participants

Two Korean children aged seven years who had visited the Yonsei University Dental Hospital participated in this study. Inclusion criteria comprised participants having ectopically erupting permanent first molar with mesial angulation of which the amount of entrapment was less than 4 mm, and those in general good health. Participants showing serious tooth mobility of primary second molar expecting an early loss and who had severely angulated ectopic molar were excluded. The consent of a child participant and parental permission was obtained from all participants to use the diagnostic records, including photographs and X-rays. All participants were fully aware of the precautions during treatment. Any management done to the participants was not harmful, and there was no serious complication. Since mRS is made of stainless steel wire, it is not toxic and hardly causes allergic reactions.

### 2.2. Modification Design

First, we changed the perpendicular angles of the horizontal part (Figure 1A) to obtuse angles (Figure 1B). This modification reduced the unwanted retracting effect of the vertical arms that interfered with safety and firm engagement (Figure 1C). The obtuse angle widened the limited distance between the two vertical arms when activated (Figure 1C) and reduced the tension on the helical spring (Figure 1D).

Second, we increased the perpendicular angles between the horizontal and vertical parts of the appliance in the lateral views (Figure 1E,F). As the increased angulation enabled the vertical arms to move more mesially, their tips could reach the mesial undercut of the locked molar with greater ease without an increase in the actual length. In addition, the swinging motion of the arms increased the distal tipping force on the locked molar (Figure 1F).

Third, we added an anterior curvature to the end of the vertical components to shorten the length of the arms (Figure 1F). Without this curvature, the ends of the two vertical arms could be entrapped in the height of contour, causing premature contact (Figure 1E). The additional anterior curvature also accelerated the swinging effect and increased stabilization during engagement by placing the ends of the arms more mesially (Figure 1D). A detailed explanation of the above-mentioned modifications is available via video online.

### 2.3. Clinical Procedure

The clinical procedure of RS or mRS is simple and only requires wiring and insertion between the ectopic erupting molar and adjacent deciduous molar under infiltrative anesthesia. Since the device produces a wedging effect, infiltrative local anesthesia on interdental papilla is required for patient discomfort control.

The distance between two vertical arms needs to be wide opened during insertion, and this can be accomplished by two utility pliers pulling each vertical arm or flattening the inner loop of the occlusal spring, which resembles the activation process. Re-activation of mRS is required every three to four weeks as the distalizing force decreases. Since mesial angulation of ectopic molar decreases as the locking improves, it is desirable to change the obtuse angle between the vertical component and horizontal component to an original perpendicular angle.

A soft food diet is recommended during treatment to prevent mRS from falling out, and patients should be careful not to swallow it.

After aligning the ectopic molar, the device is removed, and then its spontaneous eruption is awaited. Careful radiographic examination is required until the further eruption of the treated tooth to its occlusal level. We suggest a 6–12-month interval period before the follow-up to evaluate the success of treatment and complications, such as the early loss of the deciduous molar.

## 3. Case Report

Case 1: A 7-year-old boy without a history of systemic disease presented with a chief complaint of delayed eruption of the left maxillary permanent first molar. A panoramic radiograph showed 2.2 mm deep entrapment of the left maxillary first molar with considerable distal root resorption of the adjacent primary second molar; early loss of the right maxillary primary second molar and, left maxillary and mandibular primary canines, and congenitally missing right mandibular lateral incisor (Figure 2A).

We decided to unlock the left maxillary first molar using a mRS based on these clinical and radiographic evaluations. We planned the space regaining treatment of the right maxillary second premolar and left maxillary canine to take place several years later in the late mixed dentition period. The mRS was inserted after the infiltrative anesthesia (Figure 2B,C). The horizontal spring was placed on the distal cusp of the first molar without any discomfort or mucosal irritation (Figure 2C). One month after delivery, the mesial entrapment of the first molar was relieved (Figure 2D). Three months later, the maxillary left first molar was successfully unlocked and reached the same occlusal level as the adjacent primary second molar (Figure 2E). Resorption of the mesiobuccal root of the left maxillary primary second molar was observed with slightly increased mobility (Figure 2E); however, we maintained the primary second molar until its spontaneous exfoliation. The patient did not report any discomfort during this treatment. The two-year follow-up panoramic radiograph showed undisturbed full eruption of the left maxillary premolars (Figure 2F). As the patient’s left maxillary second premolar and right maxillary canine were fully erupted compared to their counterparts, we plan to initiate space regaining treatment by shifting the molars posteriorly.

Case 2: A 7-year-old girl without any relevant medical history was referred from a local clinic for her mesially tilted right maxillary first molar locked under the adjacent primary second molar. Clinical and radiographic examination revealed an approximately 2.5 mm deep entrapment of the right maxillary first molar with considerable root resorption of the right maxillary primary second molar (Figure 3A). There were no other dental abnormalities. We inserted a mRS to align the first molar after infiltrative anesthesia. The horizontal spring was placed directly above the occlusal surface of the right maxillary first molar without premature contact. The two engaging vertical arms completely embraced the mesial surface of the first molar (Figure 3B,C). One month after the insertion, a distal movement of approximately 1 mm was observed on the periapical radiograph (Figure 3D). Within 3 months, the mesially tilted right maxillary first molar escaped the entrapment, moving to its normal position with no sign of increased mobility of the right primary second molar after removal of the mRS (Figure 3E).

## 4. Discussion

Ectopic eruption of the permanent first molar presents as an irregular eruption pattern of the permanent maxillary first molar, resulting in a partial impaction under the distal surface of the primary second molar [5,10,11]. As ectopic eruption is asymptomatic, patients usually notice it after significant tooth mobility of the primary second molar occurs, or it could be accidentally found on regular radiographic examination [12]. The magnitude of impaction, severity of locking, and increased resorption of primary second molar are reliable predictive parameters of irreversible ectopic eruption [13]. In our first case, the 7-year-old boy had already lost the right maxillary primary second molar due to ectopic eruption of the right maxillary first molar when he first visited our clinic, requiring additional space regaining treatment. The outcome of ectopic eruption usually appears during age seven [6]. Routine panoramic radiographic examination at age six would be helpful for the early detection of this problem, preventing such early loss. The best space maintaining device is the deciduous tooth. According to a longitudinal study, partially resorbed primary second molars show no significant root resorption changes or repair with secondary dentin after mesial locking is relieved [14]. Resorbed deciduous molars can survive for a long time and function well regarding occlusion and space maintenance [7]. Therefore, early intervention during an asymptomatic period is critical for a favorable prognosis [2].

The treatment option for early intervention depends on the severity of impaction [2], grade of root resorption [2,10,15], degree of mesial angulation [15], and mobility of the primary second molar with the presence of subjective discomfort [10]. In mild grade with an entrapment depth of 1 mm, the interproximal wedging method using self-locking springs, such as a kesling spring, NiTi separating spring, or triangular wedging spring, could be an acceptable treatment technique [16,17]. Separators gain approximately 0.3–0.4 mm space at once, and their force generally maintains up to 1 week requiring frequent clinical visits [18]. If the depth of entrapment is over 2 mm, as in our cases, the separation method alone is insufficient, and distal tipping methods using active fixed appliances, such as Humphrey or Halterman appliance, have been traditionally recommended [19,20]. However, these appliances require additional laboratory procedures and apply an excessive anchorage burden on the adjacent primary second molar. In these two cases, we chose the mRS for unlocking the permanent first molars. It requires monthly re-activation, which is easily achieved by widening the diameter of the horizontal spring or gradually reducing the angles of the engaging arms. The unlocking procedure was completed within three months on average.

The biomechanism of the mRS is highly effective due to its scientific design. The two engaging arms exert distalizing and interproximal wedging force simultaneously. Unlike conventional fixed appliances, mRS can prevent re-locking after the termination of active distalization by separating the contact. The mesial arms also provide rigid and safe retention. As the eruption angle of the locked molar becomes steeper, the curved arms with obtuse angles enable smooth insertion following the contour of the crown and firm engagement.

However, the mRS is not universal and has several limitations. First, it is not recommended for severely tilted molars. A remarkable anterior open bite may occur after insertion due to unfavorable distal protrusion, resulting in occlusal interference. The greater the inclination of the impacted tooth, the more is the distal protrusion of the device (Figure 4). Clinicians should evaluate the radiographic view and be conscious of its use on a severely tilted molar. A self-locking spring type that has a similar function to mRS also cannot avoid occlusal interference [16]. Other methods, such as K loop or bonded NiTi wire, that are mainly placed on the buccal/lingual side of teeth, not on the occlusal surface, should be considered in such cases [21,22].

Second, it can fall out into the oral cavity and pose a risk of swallowing or aspiration. As it does not directly attach to the tooth, it can disengage, especially in an unstable mRS. If swallowed, it may be excreted after several days, and if aspirated, patients would experience frequent coughs, requiring medical consultations. To prevent aspiration, tying dental floss to the appliance during the insertion procedure is recommended, and using a mRS in patients with uncontrolled swallowing reflex should be avoided.

Third, the mRS needs to be activated frequently, every three to four weeks, owing to the high-short acting force produced by stainless steel wire. We made the helical spring as wide as possible to cover the total occlusal surface to reduce this limitation. Using beta-titanium wires with good formability and average stiffness for RS can be an alternative choice [23].

The mRS could be used as a simple and effective method to correct ectopic eruption before attempting other complicated techniques. It may succeed even in locking conditions with severe root resorption of the primary molar. The contraindications include unstable engagement and a prominent open bite due to extreme inclination. Clinicians should consider the indications and select the best appliance for the management of an ectopic eruption.

## 5. Conclusions

Irreversible ectopic eruption of the permanent first molar could cause severe root resorption and early loss of the adjacent primary second molar. Treating ectopic eruption as soon as possible before premature exfoliation of the primary second molar is essential. This clinical report describes modifications of RS and presents their advantages and limitations. The mRS does not require any laboratory procedures and can be successfully applied to various locking conditions with easy modifications. Unlocking ectopically erupting molars with mRS can be considered as the first choice of treatment.

## Figures and Tables

**Figure 1 children-08-00823-f001:**
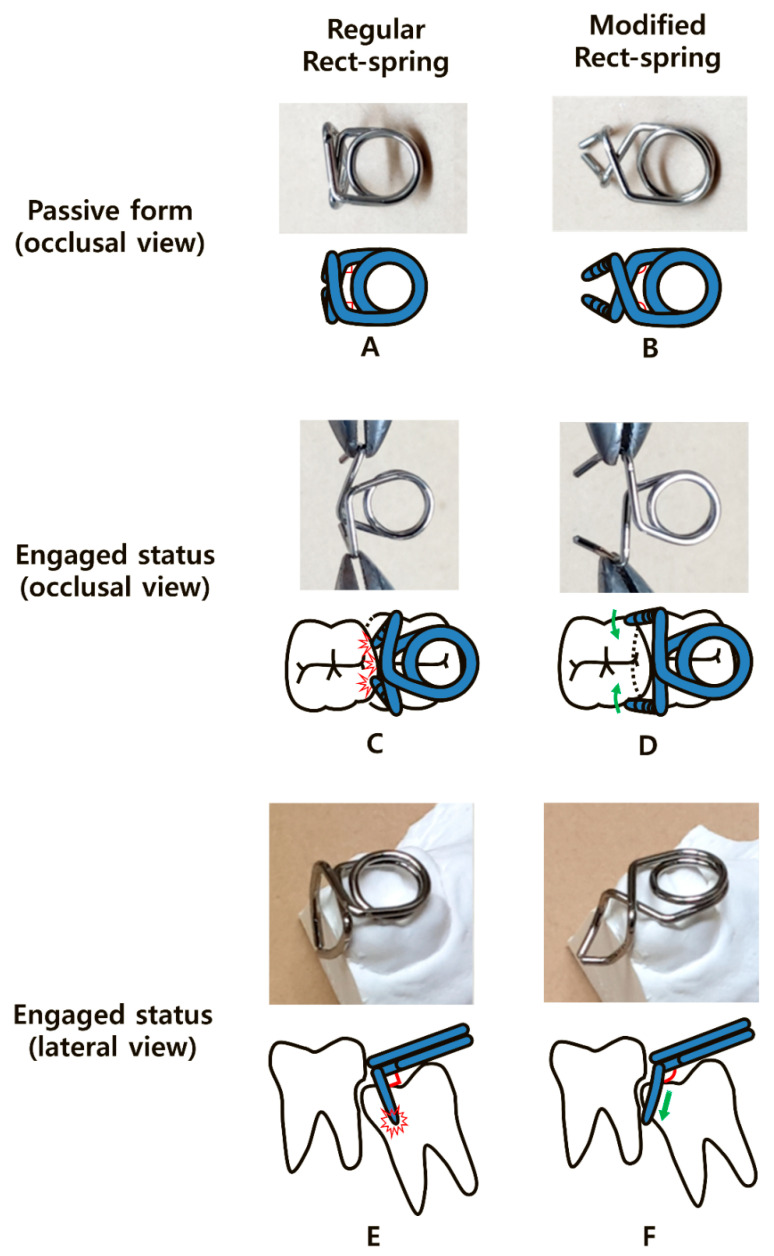
Comparison of different schematic views between the regular Rect-spring (RS) and modified Rect-spring (mRS), and the practical applications of each. (**A**) Perpendicular angles (red marks) of the RS. (**B**) Obtuse angles (red marks) of the mRS. (**C**) Unwanted retracting effect of the vertical arms during regular RS engagement, interfering with retention. (**D**) Wide opening of the vertical components of the mRS reduces the tension of the helical spring and enhances the distal movement of the locked molar on engagement. (**E**) The perpendicular angles (red angles) between the vertical and horizontal components cause unstable retention during engagement, with a risk of premature contact (red collision mark). (**F**) The obtuse angle (red angle) and curved arms engage the undercut of the locked molar (green arrow).

**Figure 2 children-08-00823-f002:**
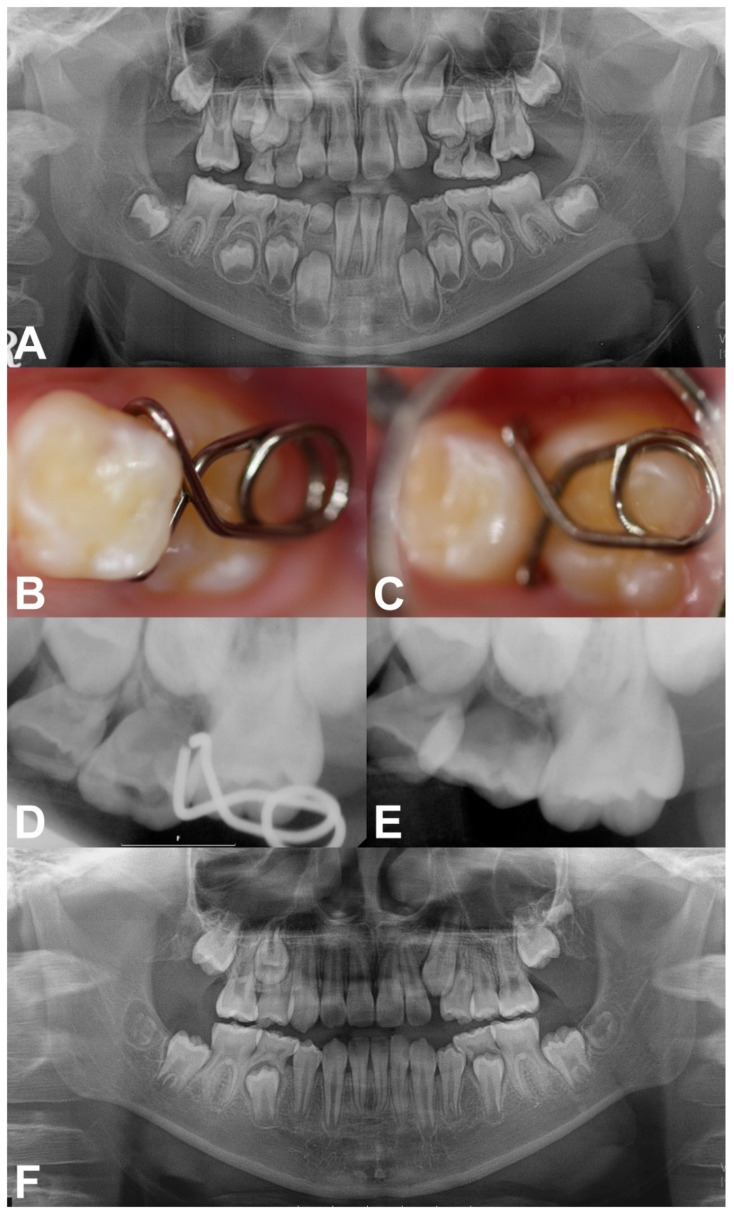
Pre-treatment and treatment progress images of case 1. (**A**) Pre-treatment panoramic radiograph at 7 years of age. (**B,C**) Clinical photograghs after the insertion of mRS. (**D**) Periapical radiograph obtained 1 month after delivery, showing relieved locking. (**E**) Post-treatment radiograph showing the successfully corrected left maxillary first molar with proper occlusion after 3 months. (**F**) Post-treatment panoramic radiograph taken at 9 years of age. Left maxillary premolars fully erupted without space deficiency.

**Figure 3 children-08-00823-f003:**
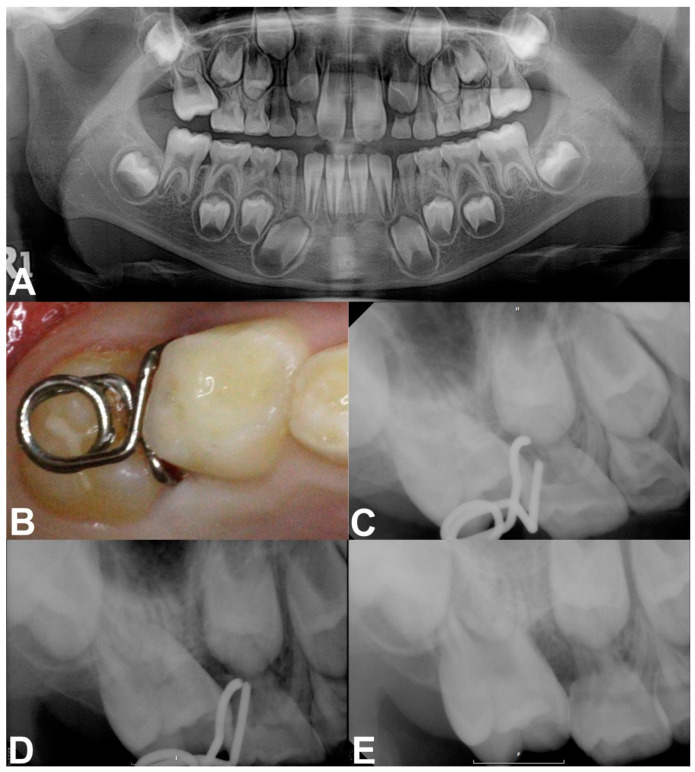
Pre-treatment and treatment progress images of case 2. (**A**) Panoramic radiograph revealed the ectopic eruption of the right maxillary first molar. (**B**,**C**) Intraoral photograph and periapical radiograph after insertion of the mRS. (**D**) Periapical radiograph obtained at the 1-month checkup. (**E**) 3 months after treatment showing a corrected right maxillary first molar.

**Figure 4 children-08-00823-f004:**
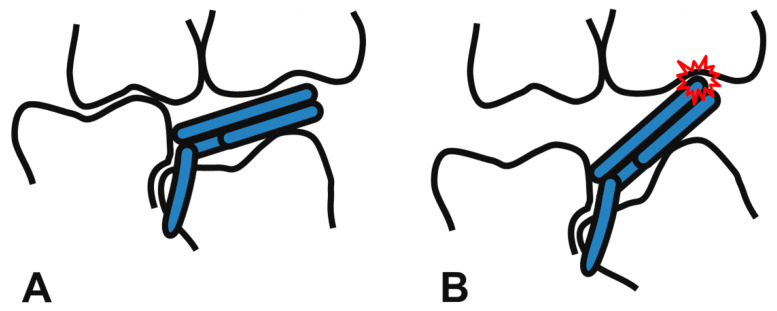
Favorable state of the mRS engaged on a slightly tilted molar (**A**), and unfavorable state of the mRS on a severely tilted molar (**B**). A severe mesial angulation of the locked molar leads to occlusal interference (red collision mark, (**B**)). RS and mRS are not recommended in such cases.

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
