# Peer review of "Modifications of Rect-Spring to Enhance the Engagement of Ectopically Entrapped Molars with 2 Case Reports"

_children, 2021, doi:10.3390/children8090823_

Round 1

Reviewer 1 Report

Dear authors, thank you for sending us your manuscript, which has been very interesting to me. However, I would like to make some considerations Despite being an article that presents two clinical cases, being an interventionist there should be a MATERIAL AND METHODS section in which the brand of the device, method of insertion and position verification, timing of spring activation, home care and control and follow-up protocol (clinical visits and evaluation of changes) should be specified.
Furthermore, the criteria for selecting patients and their ethical-legal regulations should be detailed.
Self-locking springs have a mechanism similar to the one shown in your article, that I would have liked to see in the discussion. I would like to know why it is not included in the discussion, since it is also a preformed device that relies on a controlled and small distalization of the trapped molar to allow its eruption. Thanks in advance. 

Author Response

Response 1: We agree with the reviewer’s comment, so we added MATERIAL AND METHODS section comprising information about method of insertion, position verification, timing of spring activation, home care and follow-up protocol as follows (lines 91-108):

  1. Material and Methods

2.3. Clinical Procedure

Clinical procedure of RS or mRS is simple that only requires wiring and insertion between ectopic erupting molar and adjacent deciduous molar under infiltrative anesthesia. Since the device produces wedging effect, infiltrative local anesthesia on interdental papilla is required for patient discomfort control.

The distance between two vertical arms need to be wide opened during insertion, and this can be accomplished by two utility pliers pulling each vertical arm or flattening the inner loop of the occlusal spring which resembles the activation process. Re-activation of mRS is required every three to four weeks as a distalizing force decreases. Since mesial angulation of ectopic molar decreases as the locking improves, it is desirable to change the obtuse angle between vertical component and horizontal component to an original perpendicular angle.

A soft food diet is recommended during treatment to prevent mRS not to fall out, and patients should be careful not to swallow it.

After uprighting the ectopic molar, remove the device and wait for its spontaneous eruption. Careful radiographic examination until further eruption of treated tooth to its occlusal level. We suggest a 6 – 12 month interval period of follow-up to evaluate success of treatment and complication such as early loss of deciduous molar.

However, we could not add the brand of the device on manuscript because modified Rect-spring is manually manufactured at chairside, not a ready-made device. 

Response 2: We selected two healthy patients who had ectopically erupting permanent first molar with mesial angulation. Permissions from patients and their parents were obtained. Information about criteria for selecting patients and their ethical-legal regulations that the review mentioned has been added in the material and methods, as follows (lines 56-68):

2.1.  Participants

Two Korean children aged seven years who had visited the Yonsei University Dental Hospital, participated in this study. Inclusion criteria comprised participants having ectopically erupting permanent first molar with mesial angulation of which amount of entrapment is less than 4 mm, and those in general good health. Participants showing serious tooth mobility of primary second molar expecting early loss, and who had severely angulated ectopic molar were excluded. The consent of a child participant and parental permission was obtained from all participants to use the diagnostic records including photographs and X-rays. All participants were fully aware of the precautions during treatment. Any management done to the participants was not harmful and there was no serious complication. Since mRS is made of stainless-steel wire, it is not toxic and hardly causes allergic reactions.

Response 3: Thank you for your comment. We didn’t include self-locking springs at first since the modified Rect-spring is not preformed, but rather manually manufactured at chairside only in indicated patients. Information about self-locking springs that the reviewer mentioned (new reference 16,18) has been added in the discussion, as follows (lines 185–190):

In mild grade with an entrapment depth of 1 mm, the interproximal wedging method using self locking springs such as kesling spring, NiTi separating spring, or triangular wedging spring could be an acceptable treatment technique[16,17]. Separators gain approximately 0.3 – 0.4 mm space at once, and their force generally maintain up to 1 week requiring frequent clinical visits[18].

Reviewer 2 Report

We consider that the work revised is of great clinical interest, as it is procedura that in a relatively simple and affordable way contributes to solving the frequent problem of ectopic eruption of permanent maxillary first molars.

We highlight the high quality of the images provided, wich contribute significantly to interpreting the descriptive text of the clinical cases presented.

The modifications made on the original design respond to arguments that seem to be correct. The proposed angle inlination and its wedge effect could improve both the distalization of the permanent molar and the retention of the device itself.

It would be desirable in the future to increas the number of cases, collecting in a more exhaustive and systematized way the different variables that could contribute to a better selection where the device would show its greatest effectiveness.

The revised and correctly referenced literature is very appropriate to the subject. It would be necessary to complete a publication witha difference device (Self Locking Springs type) but with a similar philosophy that avoids the occlusal helical spring and its potential interferences that we consider could be a relevant inconvenience for its use.

Best regards

Author Response

Response 1: Thank you for your comment. Based on the reviewer’s advice, we will try to overcome the shortcomings of Rect-spring continuously, and increase the number of cases using modified Rect-spring to broaden the range of patient groups where the device can show its greatest effectiveness.

Response 2: We agree with the reviewer’s comment, so we added information of different devices (new reference 16,21,22 ) that could supplement limitation of modified Rect-spring as follows (lines 210-213):

Self locking spring type which has a similar function to mRS, also cannot avoid occlusal interference[16]. Other methods such as K loop or bonded NiTi wire that are mainly placed on the buccal/lingual side of teeth, not on the occlusal surface, should be considered in such cases[21,22].
